# Effects of Essential Oil and/or Encapsulated Butyrate on Fecal Microflora in Neonatal Holstein Calves

**DOI:** 10.3390/ani13223523

**Published:** 2023-11-15

**Authors:** Zhihao Luo, Ting Liu, Peng Li, Shuru Cheng, David P. Casper

**Affiliations:** 1College of Animal Science and Technology, Gansu Agricultural University, Lanzhou 730070, China; lzhxxx0104@163.com (Z.L.); 18215133536@163.com (P.L.); chengsr@gsau.edu.cn (S.C.); 2Casper’s Calf Ranch, 4890 West Lily Creek Road, Freeport, IL 61032, USA; david.casper10@jcwifi.com; 3Department of Animal Sciences, North Carolina A&T State University, Greensboro, NC 27411, USA

**Keywords:** oregano essential oil, sodium butyrate, fecal micro-organisms, growth performance

## Abstract

**Simple Summary:**

At a young age, ruminants suffer from slow gastrointestinal development, poor digestion and metabolism, low immunity, and susceptibility to diarrhea and pneumonia. which seriously affect the healthy growth of calves and early weaning and are a key factor restricting the progress of dairy farming. The intestinal micro-organisms of ruminants will gradually stabilize with weaning and growth and no longer change easily, so young age is a critical window to regulate the intestinal micro-organisms of ruminants. Oregano essential oil and sodium butyrate have been shown to have the effect of regulating the gut microbiota of animals, and we chose these two additives as a means of regulating the gut microbiota of calves in this experiment to explore what kind of effects these two additives would have on the gut microbiota of calves.

**Abstract:**

This study was conducted to investigate the effects of feeding oregano essential oil, butyrate, and its mixture on the intestinal microbial diversity of calves. A completely randomized experimental design was used. Sixty-four healthy neonatal Holstein female calves with birth weight ≥ 35 kg were randomly divided into one control and three treatments (16 calves per group). The control group was fed normally, and the treatment group was fed oregano essential oil, butyrate, and their mixture, respectively. The experiment lasted for 70 days, and the lactation period lasted for 56 days. On days 55 and 70, rectal fecal samples from five calves were collected from each group for 16S rRNA amplification and sequencing. The results showed as follows: (1) the three treatments had no significant effects on the intestinal microbial community diversity, community uniformity, and community pedigree diversity of calves (*p* > 0.05). (2) At the phylum level, Firmicutes, Bacteroidota, Spriochatetota, Actinobacteriota, Firmicutes, and Bacteroidota gates of the main bacteria were detected in feces. (3) At the genus level, the top ten species with relative abundance detected are: *norank_ F_Muribaaculaceae*, *Ruminococcus*, *unclassified_ F_ Lachnospiraceae*, *UCG-005*, *Prevotelaceae_NK3B31_Group*, *Prevotella*, *Bacteroides*, *Rikenellaceae_RC9_Gut_Group*, and *Faecalibacterium*, *Alloprevotella*. (4) LEfSe analysis results show that the species with significant differences in the control group were *f__Lachnospiraceae*, *o__Lachnospirales*, *o__Coriobacteriales*, and *c__Coriobacteriia*, *g__Megasphaera*; in the essential oil group were *g__Lachnospiraceae_AC2044_group*, *o__Izemoplasmatales*, *g__norank_f__norank_o__Izemoplasmatales*, and *f__norank_o__Izemoplasmatales*; in the sodium butyrate group were *g__Lachnospiraceae_NK4A136_group*, *and g__Sharpea*, *g__Fournierella*; in the mixed group were *g__Flavonifractor*, *and g__UBA1819*. (5) The functional prediction analysis of calf gut microbes, found on the KEGG pathway2, shows that essential oil significantly improved membrane transport, Sodium butyrate inhibits lipid metabolism and improves the body’s resistance to disease. (*p* < 0.05). (6) The effects of each treatment on the intestinal microbial structure of calves did not last for 14 days after the treatment was stopped. In conclusion, the addition of oregano essential oil, butyrate, and its mixtures to milk fed to calves can modulate the microbial structure, and it is recommended that oregano essential oil and butyrate be used separately, as a mixture of the two can increase the rate of diarrhea in calves.

## 1. Introduction

For young ruminants, the development of the gastrointestinal tract is not only related to the survival of the animal at a young age but also to the production performance and health level of the entire life process of the animal. Studies have proved that the development of the gastrointestinal tract in ruminants is closely related to microbial colonization, and the two complement each other [1]. The gastrointestinal tract of ruminants is sterile at birth, but it will be rapidly colonized within 24 h after birth [2]. With the increase in animal age, the structure of micro-organisms will change significantly. After weaning, the changes of micro-organisms will weaken or even no longer change. Moreover, the early colonization of micro-organisms has a long-term effect on animals [3,4,5].

The micro-organisms in the rumen of ruminants mainly include bacteria, fungi, protozoa, and archaea [6], which jointly complete dietary degradation and fermentation through interaction and coordination and regulate the immune response of the body, in which bacteria play a significant role. Fungi can penetrate cell walls and enhance cellulose degradation rates by degrading lignin. In addition, *Ruminococcus flavus*, *ruminococcus albicans*, and *Filamentus succinicogenes* were the dominant groups of fibrinolytic bacteria in the rumen. The rapid growth of the three bacteria was conducive to the secretion of fibrinolytic enzymes and the digestion of dietary fiber in the rumen [7,8]. *Ruminobacter amylophilus* could utilize α-glucose as an energy source, which was crucial for starch degradation in the rumen [9]. The intensive and large-scale development of the breeding industry has brought substantial economic benefits, but at the same time, it also aggravates the bacterial infection of animals. Although antibiotics can effectively inhibit some bacterial infections, the adverse effects of antibiotic residues and bacterial resistance also pose a significant challenge to the developing animal husbandry and human health. The microbial structure is constantly changing, which is affected by variety, age, environment and diet, and the microbial system of young animals tends to be stable after adulthood. It does not change significantly [10]. Therefore, the youthful period of ruminants is the best window for regulating microbial colonization [11].

In this experiment, two new additives were selected to study whether they can play a role in the process of antibacterial and regulation of microbial colonization in ruminants. Oregano essential oil, a compound extracted from oregano, is rich in carvacrol and thymol [12], and its composition is also a critical factor in its antibacterial properties. The unique antibacterial effect of plant essential oil depends on the chemical composition it contains. The antibacterial ability of plant essential oil is related to the functional group and structural arrangement of its active molecules, and different chemical components often have synergistic antibacterial effects. Among the many active ingredients, phenols have the most potent antibacterial activity, followed by aldehydes, alcohols, ketones, esters, and hydrocarbons [13]. It has been reported that oregano essential oil can reduce the number of rumen micro-organisms, methane production, and the ratio of acetate to propionate, thereby regulating rumen Fermentation [14]. The number of *Fibrinobacter* succinate and *Ruminococcus* albums decreased, and the additive of plant essential oil could be used as a rumen fermentation regulator [15], which had an inhibitory effect on *Staphylococcus aureus* and *Streptococcus* mastitis, causing bovine mastitis [16]. The main active component of sodium butyrate is butyric acid, which is the product of microbial fermentation and is a kind of volatile fatty acid (VFA) [17], accounting for 5% to 20% of the total VFA [18]. Because butyric acid is free and volatile, it is made into a relatively stable form of sodium butyrate in production, which has a special cheese rancidity odor, is easily soluble in water and ethanol, and the pH of the aqueous solution is alkaline [19]. Studies have shown that sodium butyrate enters the stomach of animals, enters the bacterial cells with dissociation, and then decomparts into hydrogen ions and butyrate ions. When hydrogen reach a specific concentration, they will cause *E. coli*. Harmful bacteria, such as Salmonella die, while acid-resistant beneficial bacteria, such as lactic acid bacteria survive [20]. This process can maintain the dynamic balance of animal gastrointestinal micro-organisms and play a role in regulating and protecting the normal function of the gastrointestinal tract. Studies have found that sodium butyrate treatment can reduce the proportion of enterococcus and conditionally Sensitizing clostridium, and increase the balance of beneficial intestinal bacteria and lactic acid bacteria [21]. Previous studies have shown that two different substances can produce synergistic effects when used together, which can play a role in the structure of gastrointestinal microflora of animals [22]. This experiment aimed to study the effect of oregano essential oil, sodium butyrate, and their mixtures on the intestinal micro-organisms of calves, and to select the additives that would be most helpful for calf health and growth for application and in-depth study.

## 2. Materials and Methods

### 2.1. Test Materials

#### 2.1.1. Test Animals

The trial will be conducted from 12 April to 7 August 2021, at Shengyuan Breeding Base (35°39′39″ N, 103°13′50″ E) in Linxia, Gansu Province, China. The average temperature at this site during the test period was 17.4 °C, and the average humidity was 59.3% RH. Sixty-four newborn Holstein female calves were selected for the experiment, A completely randomized block group design was used, in which the newborn Holstein calves were alternately placed in the control group (CON), essential oil group (OEO), butyric acid group (CSB), and the mixed group (EPS) in the order of date of birth, with 16 calves in each group, according to the principle of random assignment. When selecting calves, individuals who were diseased and underweight at birth were excluded, and healthy individuals that were disease-free and weighed ≥35 kg at birth were assigned. Calves in the control group were fed typically. 1.25 g oregano essential oil was added to milk in the essential oil group according to dry matter calculation, 0.3% sodium butyrate was added to milk in the butyric acid group according to dry matter calculation, The mixed group added 1.25 g oregano essential oil and 0.3% sodium butyrate mixture, the addition amount of the three treatment groups remained unchanged throughout the experiment period, and the addition was stopped after weaning. Oregano essential oil was a dry granular powder containing about 1.3% oregano essential oil along with lactic acid, cobalt carbonate, and rhombohedral zeolite as a carrier purchased from Ralco Nutrition, Inc. (Marshall, MN, USA) and sodium butyrate as a 90% encapsulated form purchased from Nanchang Lifeng Industry and Trade Co. (Nanchang, Jiangxi Province, China)

#### 2.1.2. Feeding Procedure

Calves were fed 3.5–4 L of colostrum within 1 h of birth and 2 L of colostrum a second time at 6 h after birth. After that, the calves were fed with milk (acidified milk was provided on the farm where the experiment was conducted; that is, the acid mixture of methyl acid solution and water 1:9 was mixed with milk 3:100, and the nutritional composition and physical and chemical properties were shown in Table 1). On day 7, calves were transferred from the maternity ward to the island for single-pen feeding, weaned at day 56, and tested for 70 days. The feed was fed three times a day (6:00, 13:00, and 19:00) during the experiment. Calves were fed at different growth stages as follows: 1–7 days of age, three times per head, 1.5 L per feeding; 8–20 days of age, three times per day, 3 L per feeding; 21–44 days of age, three times per day, unlimited free-feeding; 45–50 days of age, twice per day, 3 L per feeding; 51–55 days of age, one time per day, 2 L per feeding. During the whole test period, milk replacer and water were freely fed. The milk replacer consisted of pellets mixed with cornflakes 5:1 (nutritional level is shown in Table 2). Pellets were purchased from Tiankang Livestock Biotechnology Co., Ltd. (Xinjiang, China).

### 2.2. Main Reagents and Instruments

E.Z.N.A.^®^ Soil DNA Kit DNA Extraction Kit (Omega Bio-tek, Norcross, GA, USA); FastPfu Polymerase (TransGen, Beijing, China); ABI GeneAmp^®^9700 PCR instrument (ABI, CA, USA); DYY-6C electrophoresis apparatus (Liuyi, Beijing, China). Illumina MiSeq high-throughput sequencing was completed by Majorbio Bio-Pharm Technology Co., Ltd. (Shanghai, China). Waters ACQUITY UPLC I-CLASS liquid chromatograph (Waters, Framingham, MA, USA); Xevo TQ-S Micro mass spectrometer (Waters, Framingham, MA, USA); BEH C8 1.7 μM 100 × 2.1 mm column.

### 2.3. Sample Collection

At 55 and 70 days of age, 5 g of fresh calf feces were collected by the rectal method, placed in a frozen storage tube, and stored in a liquid nitrogen tank to be measured.

Blood samples were collected from the jugular vein of calves at 55 days of age, and 5 mL blood samples were collected from each calf. The serum was isolated after standing at room temperature for 30 min, centrifuged at 3000 r/min for 10 min, and stored at −20 °C. After the experiment, the serum was brought back to the laboratory for determination of volatile fatty acid content.

### 2.4. Sample DNA Extraction

According to E.Z.N.A.^®^ The instructions of the soil DNA kit (Omega Bio tek, Norcross, GA, USA) were used to extract the total genomic DNA of the microbial community, 1% agarose gel electrophoresis was used to detect the quality of the extracted genomic DNA, and NanoDrop2000 (Thermo, Waltham, MA, USA) was used to determine the concentration and purity of DNA.

### 2.5. PCR Amplification and Sequencing Library Construction

Using the above extracted DNA as the template, the upstream primer 338F (5′-ACTCCTACGGGGAGGCAGCAGCAG-3′) and the downstream primer 806R (5′-GGACTACHVGGGTWTCTAAT-3′) [23] carrying the barcode sequence were used for PCR amplification of the variable region of 16S rRNA gene V3–V4. The amplification procedure was as follows: pre-denatured at 95 °C for 3 min, 27 cycles (denatured at 95 °C for 30 s, annealed at 55 °C for 30 s, extended at 72 °C for 45 s), and then stably extended at 72 °C for 10 min, Finally, store it at 4 °C (PCR instrument: ABI GeneAmp^®^ 9700, ABI, CA, USA). PCR reaction system: 5 × TransStart FastPfu Buffer 4 μL, 2.5 mM dNTPs 2 μ 50. Upstream primer (5 μM) 0.8 μ 50. Downstream primer (5 μM) 0.8 μ 50. TransStart FastPfu DNA Polymerase 0.4 μ 50. Template DNA 10 ng, supplemented to 20 μL. Three replicates per sample. Mix the PCR products of the same sample, use 2% agarose gel to recover the PCR products, use the AxyPrep DNA Gel Extraction Kit (Axygen Biosciences, Union City, CA, USA) to purify the recovered products, use 2% agarose gel electrophoresis to detect, and use the Quantum ™ Fluorometer (Promega, Madison, WI, USA) detects and quantifies the recovered products.

Use NEXTFLEX Rapid DNA Seq Kit to build a library of purified PCR products: (1) connector link, (2) magnetic bead screening was used to remove the self-connected segment of the joint, (3) PCR amplification was used to enrich the library template, (4) The final library was obtained by recovering PCR products from magnetic beads. Sequencing was performed using the Miseq PE300 platform of Illumina (Shanghai Meiji Biomedical Technology Co., Ltd., Shanghai, China).

### 2.6. High-Throughput Sequencing Data Analysis

Use fastp [24] (version 0.19.6) software (version 0.19.6); use FLASH [25] Software (version 1.2.11) splicing: (1) Filter the bases with the tail mass value of reads below 20, set a 50 bp window; if the average mass value in the window is lower than 20, cut off the back base from the window, filter the reads with the tail mass value below 50 bp after quality control, and remove the reads containing N base. (2) According to the overlap relationship between PE reads, merge the paired reads into a sequence, with the minimum overlap length of 10 bp. (3) The maximum mismatch ratio allowed in the overlap region of spliced sequences was 0.2, and nonconformant sequences were screened. (4) Samples are distinguished according to the barcodes and primers at both ends of the sequence, and the sequence direction is adjusted. The allowable mismatch number of barcodes is 0, and the maximum primer mismatch number is 2. Based on the default parameters, use the DADA2 [26] plug-in in Qiime2 process [27] to reduce the noise of the optimized sequence after QC splicing. The sequences after DADA2 noise reduction processing are usually called ASVs (i.e., amplified subsequence variants). Remove the chloroplast and mitochondrial sequences annotated in all samples. In order to minimize the impact of sequencing depth on subsequent alpha diversity and beta diversity data analysis, the number of all sample sequences is leveled according to the minimum sample sequence number. After leveling, the average sequence coverage of each sample can still reach 99.90%. Based on the Sliva 16S rRNA gene database (v. 138), the Naive Bayes classifier in Qiime2 was used to conduct species taxonomic analysis of ASVs.

### 2.7. Volatile Fatty Acid Detection

The serum sample was melted at room temperature, vortexed, and mixed. Approximately 50 µL was taken into a labeled 1.5 mL centrifuge tube, and 150 µL of methanol was added, vortexed, and shaken for 20 min. 12,000 rpm, 4 degrees Celsius, centrifuged for 5 min, 50 µL of the upper layer of the solution was taken and derivatized with derivatization reagent. After derivatization was completed, it was put into a 2 mL injection vial with a 200 µL liner, and then assayed and measured on a machine.

### 2.8. Statistical Analysis

All data analysis is conducted on the Meiji Biological Cloud Platform. The details are as follows: Mothur [28] software (version 1.30) is used. Calculate the alpha diversity knowledge Chao 1, Shannon, PD, Simpsoneven index, and use Wilxocon rank sum test to analyze the differences between groups of alpha diversity. PCoA analysis (principal coordinate analysis) based on the Bray-Curtis distance algorithm was used to test the similarity of microbial community structure among the samples, and a PERMANOVA nonparametric test was used to analyze whether there was a significant difference in microbial community structure among the sample groups; linear discriminant analysis Effect Size [29] (LDA > 3, *p* < 0.05) determines the influence of bacterial community structure with a significant difference from phyla to genus level abundance among different groups. A correlation between two nodes was considered to be statistically robust if the Spearman’s correlation coefficient was over 0.6 or less than −0.6, and the *p*-value less than 0.01; sequence functional abundance was predicted using PICRUSt2 [30] (v2.2.0-b); the volatile fatty acid data were analyzed by one-way ANOVA using IBM SPSS Statistics 25, with different treatments as the only variable factor and with *p* < 0.05 as the criterion for significant differences.

## 3. Results

### 3.1. Sequencing Results and Diversity Analysis

#### 3.1.1. Basic Sequencing Results

The diversity data of 40 samples were analyzed, and a total of 2,154,177,893,215,123 bases were obtained. Statistics of species annotation results: Domain: 1 Kingdom: 1 Phylum: 13 Class: 21 Order: 45 Family: 77 Genus: 223 species: 445 ASV: 2625.

#### 3.1.2. Dilution Curve

The dilution curve can directly reflect the rationality of the amount of sequencing data and indirectly reflect the richness of species in the samples. When the curve tends to be flat, it indicates a reasonable amount of sequencing data. As shown in Figure 1, with the increase in adequate sequence sequencing depth, the dilution curve showed a sharp rise at first and then gradually leveled off. This change trend indicated that the sequencing data of the samples in this experiment was reasonable, the sequencing quality was good, and the samples had a certain depth and representativeness.

#### 3.1.3. Analysis of Species Composition

To obtain the species classification information corresponding to each ASV, the classify-sklearn (Naive Bayes) algorithm is used to classify the representative sequence of ASV. Species annotation database: silva138/16s_bacteria confidence: 0.7. Obtain annotation information for ASV at different classification levels. A total of 1 domain, 1 kingdom, 13 phyla, 21 classes, 45 orders, 77 families, 223 genera, and 445 species were obtained. According to the Venn diagram (Figure 2), 1848 ASVs were detected in the intestines of calves before weaning, of which 220 were unique in CON, 304 in OEO, and 181 in CSB. There were 232 ASVs in EPS, and 1871 ASVs were detected in the intestines of calves after weaning, including 260 ASVs in CON, 248 ASVs in OEO, 220 ASVs in CSB, and 183 ASVs in EPS.

#### 3.1.4. Alpha Diversity Analysis

Alpha diversity is used to analyze microbial community diversity within a sample. Chao1 reflects community richness. Shannon reflects community diversity. Simpson even reflects community evenness. PD reflects community lineage diversity. As can be seen from Figure 3, there were no differences in the four indexes among the groups before and after weaning.

#### 3.1.5. Beta Diversity Analysis

Beta diversity mainly analyzes the similarity of microbial community composition between samples. Principal coordinate analysis was used to analyze the beta diversity of samples in this experiment. As can be seen from Figure 4, there was no significant difference in fecal microbial structure composition between the two groups. Based on the Bray–Curtis algorithm, Adonis analysis showed no significant differences between and within the groups of micro-organisms.

### 3.2. Species Composition Analysis at the Phylum and Genus Level

In this study, the top 10 species with maximum abundance were analyzed at two taxonomic levels, phylum and genus.

Phylum level: As can be seen in Figure 5A, Firmicutes, Bacteroidota, Spriochatetota, and Actinobacteriota were detected before weaning, where Firmicutes and Bacteroidota were the significant phyla detected in feces, and Firmicutes, Bacteroidota, Spriochatetota, Actinobacteriota, and Proteobacteria, where Firmicutes and Bacteroidota were the major phyla detected in feces (Figure 5A), and after weaning. Bacteroidota, Spriochatetota, Actinobacteriota, Proteobacteria, where Firmicutes and Bacteroidota were the significant phyla detected in feces (Figure 5C), and the relative abundance of phylum-level species in each group and the ratio of the phylum Actinobacteria to the thick-walled phylum (F/B) in each group are shown in the table below. Ratio (F/B) is shown in Table 3.

Genus Level: As can be seen in Figure 5B, the top ten genera detected in relative abundance before weaning were mainly: *norank__f__Muribaculaceae*; *Ruminococcus*; *unclassified__f__Lachnospiraceae*; *UCG-005*; *Prevotellaceae_NK3B31_ group*; *Prevotella*; *Bacteroides*; *Rikenellaceae_RC9_gut_group*; *Faecalibacterium*; *and Alloprevotella*. The top ten species detected in relative abundance after weaning were (Figure 5D): *norank__f__Muribaculaceae*; *UCG-005*; *Prevotellaceae_NK3B31_group*; *unclassified__f__Lachnospiraceae*; *Prevotella*; *Christensenellaceae_R7_group*; *Ruminococcus*; *Rikenellaceae_RC9_gut_group*; *Alloprevotella*; *Blautia*, and the relative abundance of genus-level species in each group is shown in Table 4.

### 3.3. Species Difference Analysis

Linear discriminant analysis was used to analyze the differences between multilevel species (Figure 6A), trying to find the species with significant differences in different groups of dairy cows. Combined with the LDA effect value and *p*-value, the influence of fecal microbial species abundance on the difference effect was compared, and the different species were screened out (LDA score ≥ 3, *p* < 0.05). Results show that: Before weaning, the relative abundance of *f__Lachnospiraceae*, *o__Lachnospirales*, *o__Coriobacteriales*, *c__Coriobacteriia* and *g__Megasphaera* in CON increased significantly. The relative abundance of *g__Lachnospiraceae_AC2044_group*, *o__Izemoplasmatales*, *g__norank_f__norank_o__Izemoplasmatales* and *f__norank_o__Izemoplasmata* in OEO increased significantly, the relative abundance of *g__Lachnospiraceae_NK4A136_group*, *g__Sharpea*, *g__Fournierella* in CSB increased significantly. The relative abundance of *g__Flavonifractor*, *g__UBA1819* in EPS increased significantly. After weaning, the relative abundance of *o__Clostridia_vadinBB60_group*, *f__norank_o__Clostridia_vadinBB60_groupc*, *g__norank_f__norank_o__Clostridia_vadinBB60_group* and *g__UCG-004* in EPS increased significantly.

### 3.4. Microbial Function Prediction Analysis

Predictions of calf gut microbial function showed (in the KEGG level 2 pathway) that it was the same between treatment groups before and after weaning (Figure 7). The main functions of calf gut microbes (top 10 in functional abundance) were global and overview maps, carbohydrate metabolism, amino acid metabolism, energy metabolism, metabolism of cofactors and vitamins, translation, replication and repair, membrane transport, nucleotide metabolism, signal transduction, and the functional enrichment in each group is shown in Table 5.

In the KEGG level 2 pathway (Figure 8), OEO significantly increased the membrane transport pathway before weaning (*p* < 0.05), and CSB significantly increased the lipid metabolism pathway, and Infectious disease: viral pathway after weaning. (*p* < 0.05).

### 3.5. Correlation Analysis of Gut Microbiome and Production of Volatile Fatty Acids in Calves

As shown in Figure 9, species that were significantly correlated with acetic acid content (*p* < 0.05) were *unclassified_f__Oscillospiraceae*, and species that were significantly correlated with propionic acid content (*p* < 0.05) were *Subdoligranulum*; *Oscillospira*, the species that were significantly correlated with isobutyric acid content (*p* < 0.05) were *Phascolarctobacterium*, *Lachnospiraceae_AC2044_group*, *Family_XIII_AD3011_group*, and *norank_f__norank_o__Gastranaerophilale. norank_f__norank_o__Gastranaerophilale* was significantly correlated with isobutyric acid content (*p* < 0.01). The species *norank_f__norank_o__RF39* was significantly correlated with butyric acid content (*p* < 0.05). *Olsenella; Subdoligranulum*; among *Oscillospira*, *norank_f__norank_o__RF39* had a very significant correlation with butyric acid content (*p* < 0.01), and *Monoglobus* had a significant correlation with valeric acid content (*p* < 0.05). *unclassified_f__Oscillospiraceae; norank_f__norank_o__RF39*; *Olsenella*; *Subdoligranulum*; *Roseburia*, in which *Subdoligranulum* was significantly correlated with valeric acid content (*p* < 0.01), and *Blautia* was significantly correlated with isovaleric acid content (*p* < 0.05).

## 4. Discussion

### 4.1. Effects of Oregano Essential Oil, Sodium Butyrate and Their Mixtures on Intestinal Microbial Diversity of Calves

Microbial community diversity is explained by the Alpha diversity index and Beta diversity analysis (PCoA analysis). Alpha diversity is mainly used to study community diversity in a certain sample, and information such as species richness and diversity in environmental communities can be obtained by evaluating a series of Alpha diversity indexes. Four indices, Chao, Shannon, Simpsoneven and PD, were used in this experiment. The results showed no difference between the OEO, CSB, and EPS compared with CON, indicating that the three treatments had no significant effects on the richness, diversity, and uniformity of microbial communities. Beta diversity can be used to study the similarities or differences in sample community composition. The results of PCoA analysis showed that the microflora of each group could not be significantly separated, indicating that the microflora of each component did not have high stability and had no significant differences in microbial structure.

### 4.2. Effects of Oregano Essential Oil, Sodium Butyrate, and Their Mixtures on Intestinal Microbial Species Composition of Calves

At the phylum level, Firmicutes and Bacteroidota, the top two species with relative abundance, were detected before and after weaning. This is consistent with the results reported in previous studies [31]. It is the core bacteria of the mammalian gastrointestinal tract [32]. Firmicutes, Bacteroidota, Spriocatetota, and Actinobacteriota, were detected before weaning. There was no significant difference in abundance among the groups. Proteobacteria with high abundance were detected after weaning. Proteobacteria exist widely in soil, sewage, and human and animal feces and are conditional pathogens. When body resistance decreases, it can cause various infections, such as urinary tract infections and respiratory tract infections. According to research reports, proteobacteria may take advantage of host defense, drive proinflammatory changes, and alter the intestinal microbiota, leading to microbial ecological disorders in animals [33]. At the genus level, the species with the highest relative abundance was *norank_f__Muribaculaceae*, and relevant reports showed that it was positively correlated with nose-site PH and milk yield, and could reduce the abundance of spirochaetes [34]. The second most abundant species is *Ruminococcus*, which has the highest abundance in CSB. *Ruminococcus* can inhibit the expression of proapoptotic genes, inflammatory genes, and heat shock proteins [35]. *Lachnospiraceae* is a micro-organism that can produce n-butyrate in the intestine, which may be related to host energy regulation and intestinal mucosal integrity [36]. *UCG-005* was positively correlated with butyrate production [37], *Prevotellaceae_NK3B31_group* was associated with butyrate production [38]. The abundance of all three treatment groups was higher than CON, and the highest abundance was found in EPS. *Prevotella* helps to improve intestinal immunity [39]. Only OEO had a higher abundance than CON. The abundance of *Bacteroides* was positively associated with the abundance of *Bacteroides* was positively correlated with the colonization resistance of intestinal micro-organisms [40], which can inhibit the invasion of pathogens in the early stage of infection and also plays an essential role in energy production and metabolism of the animal body [41]. Its abundance did not change between OEO, CSBand CON, but decreased in EPS, which may be caused by antagonism between the two additives. *Rikenellaceae_RC9_gut_group* is positively correlated with volatile fatty acids, which may regulate the deposition of muscle fat by influencing the concentration of volatile fatty acids [42]. *Rikenellaceae_rc9_gut_group* has intestinal protective function [43] and fiber decomposition ability [44], and its abundance in CSB is higher thanCON. *Faecalibacterium* is an essential butyric acid-producing bacterium [45] that can provide energy for the body and reduce inflammation. Its increase in content is closely related to the intestinal health of the host [46]. Its abundance increases in OEO and decreases in CSB and EPS, indicating that butyric acid is negatively correlated with its mass. *Alloprevotella* can improve fiber digestibility [47]. Compared with CON, the mass of OEO and CSB did not change, while the mass of EPS increased.

### 4.3. Effects of Oregano Essential Oil, Sodium Butyrate, and Their Mixtures in Species Differences of Intestinal Microbiome in Calves

LEfSe analysis represents biomarkers with statistical differences between groups. That is, species with significant differences between groups can be represented by the LDA score. In this experiment, the LDA score distribution histogram was drawn with LEfSe analysis. When the LDA score was greater than the set value 3, species of OEO with significant differences from other groups include *Lachnospiraceae_AC2044_group*, *Izemoplasmatales*, and *Lachnospiraceae_AC2044_group*, which are important members of Trichospiraceae. *Lachnospiraceae_AC2044_group* has a significant effect on animal glucose metabolism, especially polysaccharide degradation. It has been reported that the relative abundance of *Lachnospiraceae_AC2044_group* is negatively correlated with cholesterol level, which may have an important impact on the health of calves [48]. They can also serve as target bacteria to regulate dietary rumen fermentation [49]. There are few reports about *Izemoplasmatales* that need further study. The species in CSB showed significant differences from the other groups, including *g__Lachnospiraceae_NK4A136_group*, *g__Lachnospiraceae_UCG-003*, *g__Coprobacter*, *g__Fournierella*, and *g__Sharpea Lachnospiraceae_NK4A136_group* and have been reported to repair the mucosal barrier and effectively relieve colitis [50]. Fournierella has been reported to be positively correlated with animal feed conversion and average daily intake [51]. Sharpea has been reported to relieve diarrhea in animals [52]. Species of EPS with significant differences compared with other groups include *g__UBA1819* and *g__flavonractor*, which is reported as a positive correlation between the relative abundance of intestinal *UBA1819* in calves and the occurrence of diarrhea [53]. It is reported that *Flavonifractor* is related to short-chain fatty acid production [54] and can improve animal growth performance.

### 4.4. Effects of Oregano Essential Oil, Sodium Butyrate, and Their Mixtures on Intestinal Microbes of Calves

It has been reported that oregano essential oil can alter many of the bacterial genera associated with feed digestibility by modulating fermentation [55]. It can also increase the concentration of short-chain fatty acids [56]. As seen in Figure 10, OEO significantly increased the relative abundance of *Lachnospiraceae_AC2044_group* and *norank_f__norank__o___Izemoplasmatales* in this experiment. This change in abundance could promote glucose metabolism and reduce cholesterol levels in the animal body. *Lachnospiraceae_AC2044_group* was significantly positively correlated with isobutyric acid content. And according to Table 6, OEO has the highest content of isobutyric, acetic, butyric, valeric, and isovaleric acids. In the functional prediction analysis, it was found that the membrane transporter pathway significantly increased in OEO. Adding oregano essential oil could promote the digestion and absorption of nutrients and the formation of short-chain fatty acids in the organism, which was in line with the previous reports.

Some studies have reported that sodium butyrate can improve growth performance and enhance the intestinal mucosal barrier function of livestock and poultry by changing the intestinal flora [57]. In this experiment, CSB significantly increased the relative abundance of *Lachnospiraceae_NK4A136_group*, *Fournierella*, *Sharpea*, *Coprobacter*, and *Lachnospiraceae_UCG-003*, and this change in quantity could alleviate gastrointestinal inflammation, improve feed conversion rate, and alleviate calf diarrhea. CSB had the lowest F/B among the four groups, with a decrease in the F/B ratio and an increase in the relative abundance of Lactobacillus, which would exhibit immunomodulatory effects by inducing T-cell polarization and increasing *Lactobacillus* in the intestinal flora [58], which is in agreement with previous reports.

EPS significantly increased the relative abundance of *g__UBA1819* and *g__Flavonifractor*, and this change in quantity could promote the formation of short-chain fatty acids, which could encourage calf digestibility. Still, the increase in the abundance of *g__UBA1819* also led to a rise in the probability of calf diarrhea. The reason for this is that when oregano essential oil and sodium butyrate are added together, the high concentration of the additive burdens the calf’s gastrointestinal tract, resulting in damage to the gastrointestinal tract and thus diarrhea, which can be investigated by lowering the attention of the two additives in subsequent studies.

### 4.5. Duration of Influence of Treatment Group on Microbial Structure

Weaning and treatment were stopped on the 56th day of the feeding period. Fecal samples were collected and sequenced again on the 70th day of the feeding period, two weeks after the supplementation of oregano essential oil and butyric acid was stopped, to explore the duration of the effects of each treatment group on the gastrointestinal microstructures of calves. Microbial composition, species abundance, species differentiation, and function were all found to be altered two weeks after cessation of the treatments, which led to the conclusion that the treatments of oregano essential oil, butyric acid, and a mixture of the two lasted less than two weeks after cessation. It is suggested that subsequent studies could be conducted at shorter intervals, which would allow for further investigation of the duration of the effects of the different treatments on the microbial structure.

## 5. Conclusions

The present study concluded that feeding oregano essential oil, butyric acid, and a mixture of the two to calves at a young age did not significantly affect intestinal microbial diversity, but significantly affected the relative abundance of some species, which resulted in significant effects on feed digestion, fatty acid formation, protection of intestinal micro-ecology, and calf immunity. The effect of the oregano essential oil group is mainly reflected in the promotion of calf digestion and absorption ability. The effect of the sodium butyrate group is reflected in the promotion of digestion, increased feed intake, reduced rate of diarrhea, and enhanced immune ability. The mixed group can promote the formation of short-chain fatty acids, but, at the same time, also exacerbate the probability of diarrhea appearance, so it is not recommended to mix oregano essential oil with sodium butyrate under the conditions of this study. In addition, by detecting microbial communities two weeks after stopping the treatments, it was found that none of the three treatments could maintain their effects up to 14 days after stopping, and the present study provides theoretical support for the use of novel additives in animal production, and also lays the basis for subsequent research.

## Figures and Tables

**Figure 1 animals-13-03523-f001:**
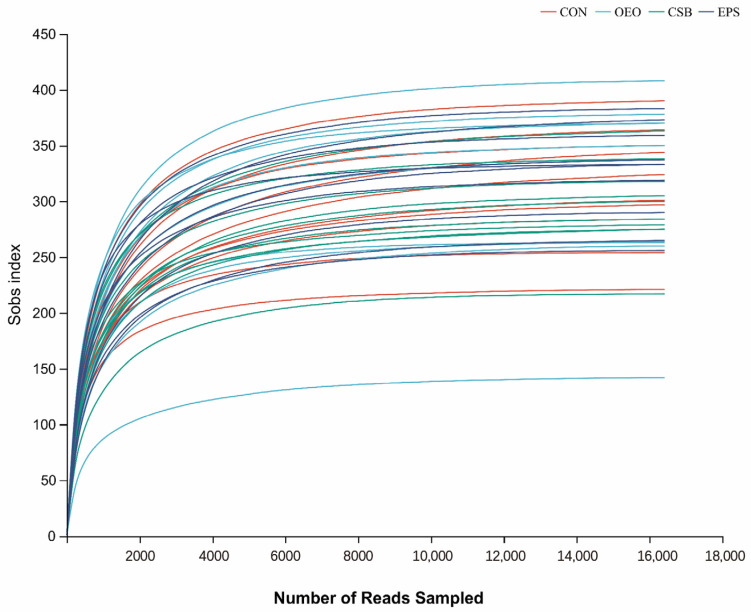
Species dilution curve. The different colored lines in the upper right corner represent different subgroups, and the horizontal coordinate is the amount of randomly selected sequencing data; the vertical coordinate is the number of species observed.

**Figure 2 animals-13-03523-f002:**
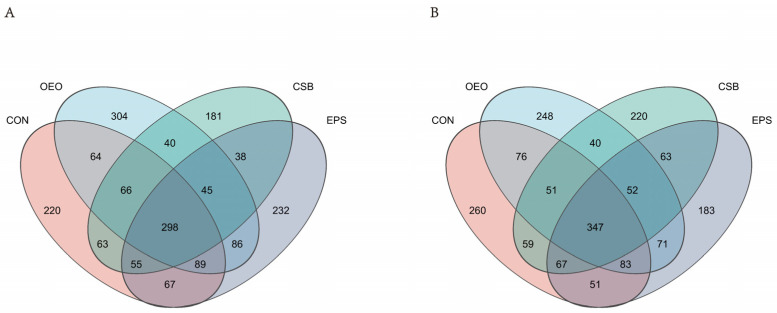
Venn diagram of the species composition of intestinal micro-organisms in calves. (**A**) Venn diagram analysis of calves before weaning. (**B**) Venn diagram analysis of calves before weaning. Different colors represent different groups, overlapping parts represent genus that are common among multiple groups, parts of not overlap represent genus that are specific to the group, and numbers indicate the number of genus corresponding.

**Figure 3 animals-13-03523-f003:**
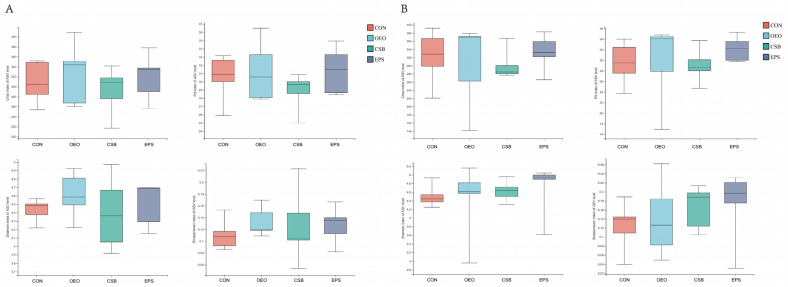
Box diagram of α diversity analysis. (**A**) Analysis of alpha diversity of calves before weaning, from left, Chao, PD, Shannon, and Shannoneven indices. (**B**) Analysis of alpha diversity of calves before weaning, same order as above. This graph demonstrates the significant differences between the two selected sample groups, with the group names in the horizontal coordinates and the exponential means for each group in the vertical coordinates.

**Figure 4 animals-13-03523-f004:**
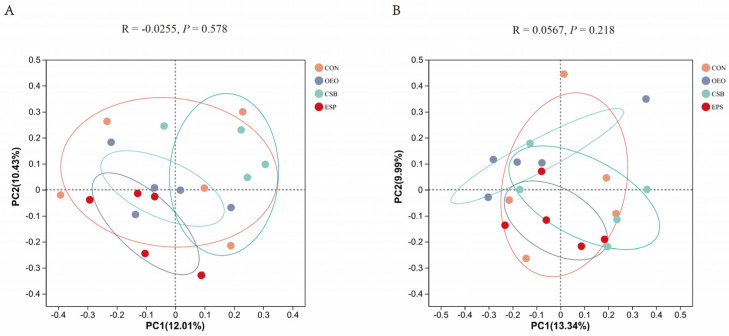
PCoA principal coordinate analysis. (**A**) PCoA analysis chart of calves before weaning. (**B**) PCoA analysis chart of calves after weaning. The horizontal and vertical coordinates represent the two selected principal coordinate components, and the percentage represents the contribution of the principal coordinate components to the differences in sample composition. The closer the R-value is to 1, the greater the between-group difference is than the within-group difference, and the smaller the R-value is, the less significant the between- and within-group differences are.

**Figure 5 animals-13-03523-f005:**
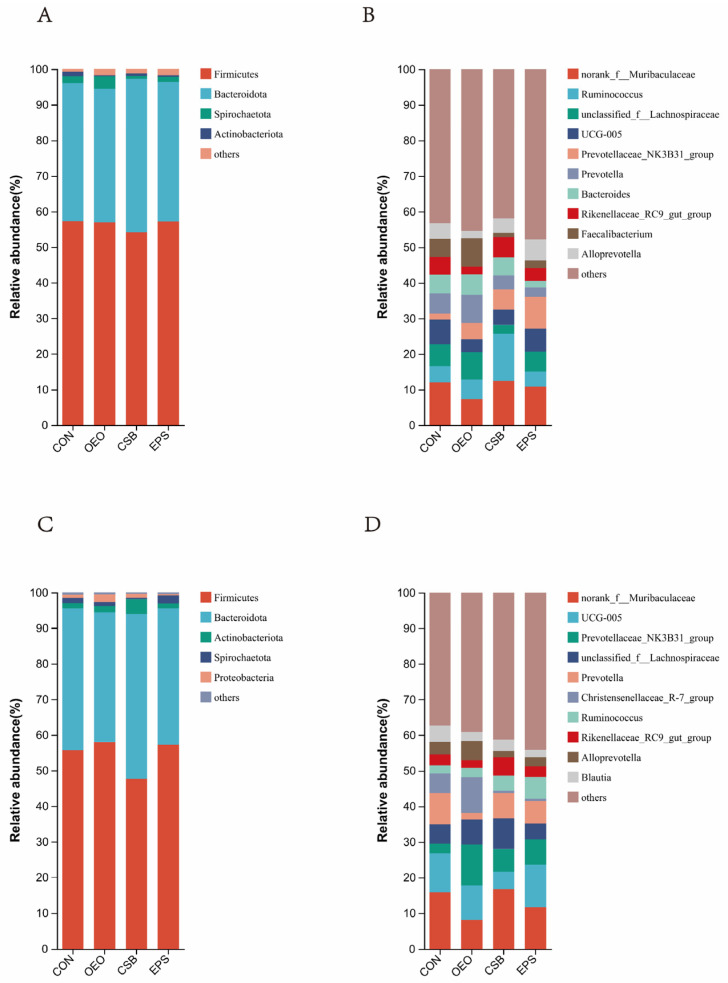
Species bar chart at phylum and genus level. (**A**) Species bar chart of calves at phylum level before weaning. (**B**) Species bar chart of calves at genus level before weaning. (**C**) Species bar chart of calves at phylum level after weaning. (**D**) Species bar chart of calves at genus level after weaning. The horizontal coordinate is the name of the grouping, the vertical coordinate is the proportion of the species in that sample, different colored bars represent different species, and the length of the bar represents the size of the proportion of that species.

**Figure 6 animals-13-03523-f006:**
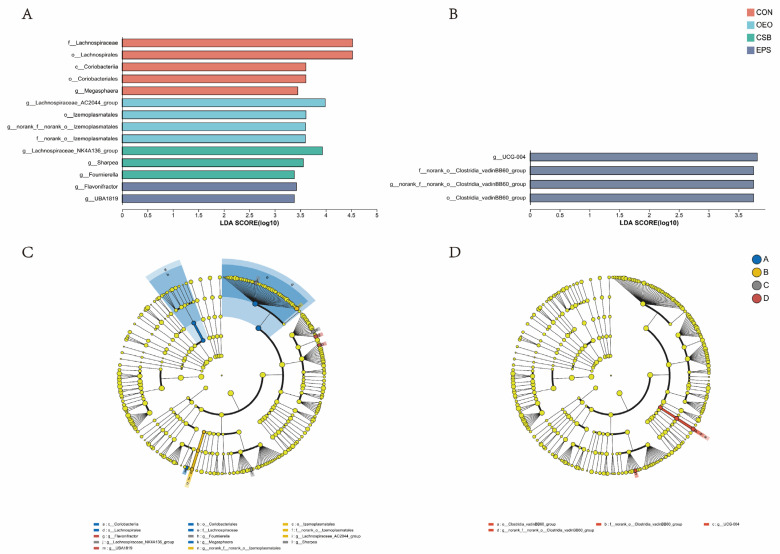
Species differences Lefse analysis plot. (**A**) Histogram of LDA discrimination in calves before weaning. (**B**) Histogram of LDA discrimination in calves after weaning This figure counts the microbial taxa with significant effects in multiple groups, and LDA scores were obtained by LDA analysis, with larger LDA scores representing greater effects of species abundance on differential effects. (**C**) Cladogram of calves before weaning. (**D**) Cladogram of calves after weaning. Different colored nodes indicate microbial taxa that are significantly enriched in the corresponding group and have a significant effect on intergroup differences; yellowish nodes indicate microbial taxa that are not significantly different in any of the different subgroups or have no significant effect on intergroup differences.

**Figure 7 animals-13-03523-f007:**
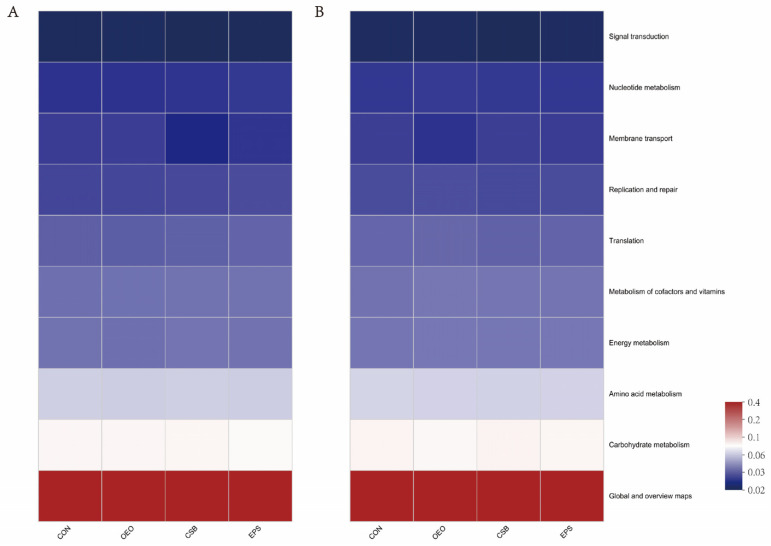
Prediction of intestinal microbial function of calves in the KEGG pathway 2. (**A**) Functional Heatmap of calves before weaning. (**B**) Functional Heatmap of calves after weaning. The horizontal coordinate is the group name, the vertical coordinate is the MetaCyc pathway function name, and the color gradient of the color block is used to show the change of abundance of different functions in the sample/group, and the legend shows the values represented by the color gradient.

**Figure 8 animals-13-03523-f008:**
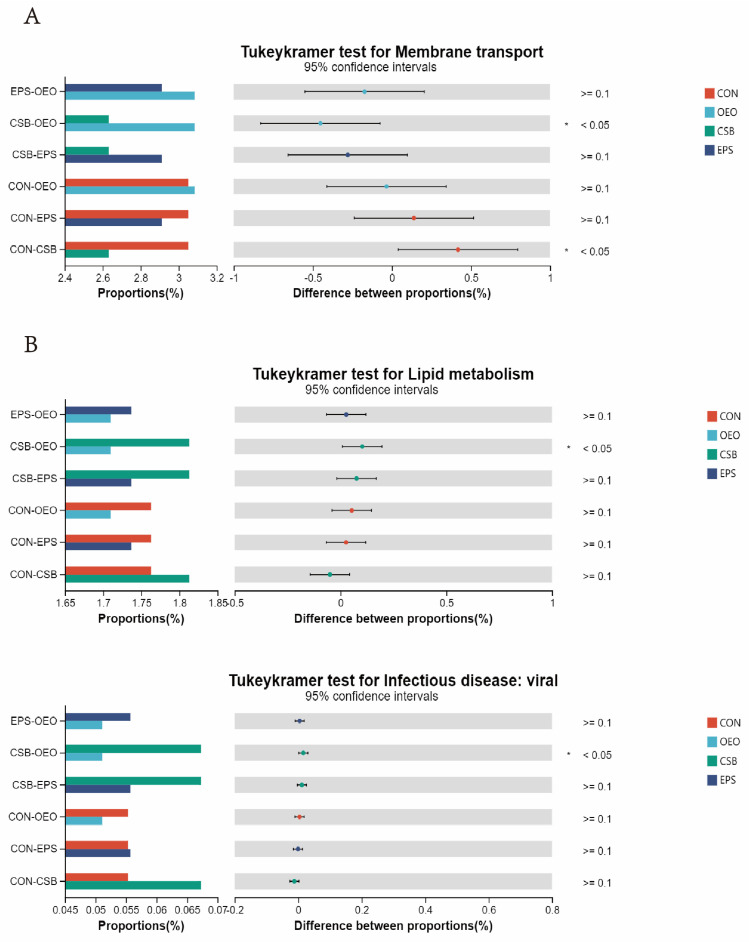
Differences in intestinal microbial function of calves in the KEGG pathway 2. (**A**) Diagram of functional differences of calves before weaning. (**B**) Diagram of functional differences of calves after weaning. The X-axis of the bar chart on the left indicates the mean relative abundance of a species in different subgroups, the vertical coordinate indicates the subgroup category for two-by-two comparisons in multiple groups, different colors indicate different subgroups, and the *p*-value is shown on the right, * 0.01 < *p* < 0.05.

**Figure 9 animals-13-03523-f009:**
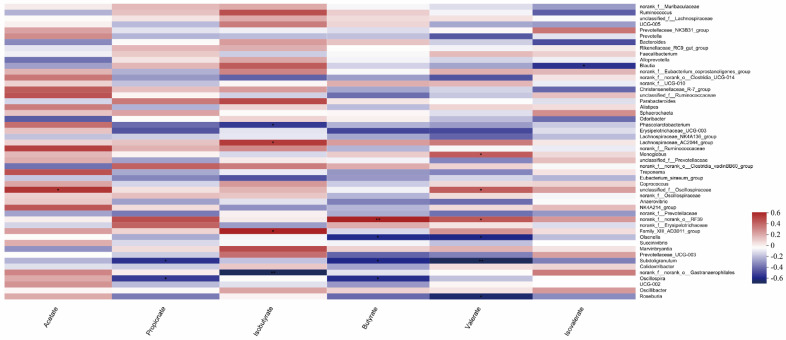
Correlation analysis of gut microbes and volatile fatty acids in calves (Note: The *X*-axis and Y-axis are environmental factors and species, respectively, and R and *p*-values are obtained by calculation. R-values are shown in different colors in the graph. If the *p*-value is less than 0.05, it is marked with a *. The legend on the right is the color interval of different R-values. You can choose to present a cluster tree of species and environmental factors (such as left and top). ** 0.001 * 0.01 < 0.05, or less *p* < 0.01).

**Figure 10 animals-13-03523-f010:**
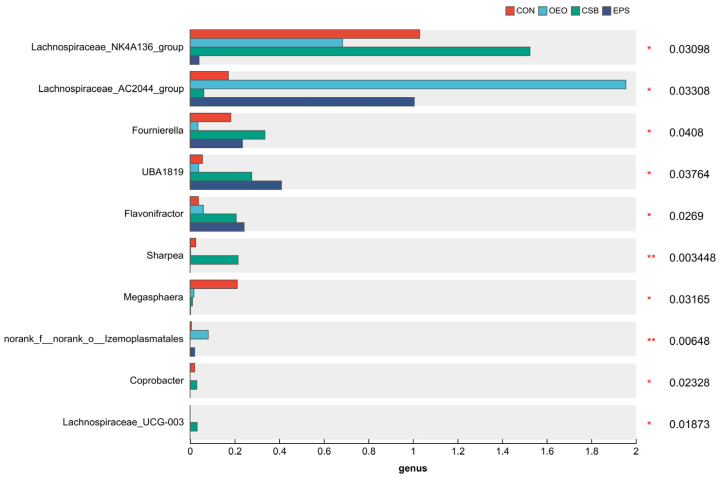
Gut microbial species of 55-day-old calves (at genus level). Horizontal coordinates indicate species names at different taxonomic levels, vertical coordinates indicate the percentage value of abundance of a species for that sample, and different colors indicate different groupings. The rightmost is the *p*-value, * 0.01 < *p* ≤ 0.05, ** 0.001 < *p* ≤ 0.01.

**Table 1 animals-13-03523-t001:** Nutrient composition and physicochemical indexes of acidified milk.

Items	Acidified Milk
Milk fat (%)	2.81
Milk protein (%)	3.29
SNF (%)	9.58
Lactose (%)	4.88
Ash (%)	0.74
TDM (%)	11.86
Proportion	31.63
pH	4.13
Conductivity (mS/cm)	7.72
Freezing Point (°C)	−0.558

**Table 2 animals-13-03523-t002:** Composition and nutrient levels of milk replacer (Air dried basis) %.

Ingredients	Content	Nutrient Level	Content
Corn	40.54	DM	87.95
Soybean meal	32.00	CP	22.17
Wheat bran	5.80	EE	3.79
Cottonseed meal	5.30	Ash	5.91
Puffed soybeans	5.00	ADF	6.18
Whey powder	4.00	NDF	12.23
Molasses	4.00	Ca	0.91
CaCO_3_	1.60	P	0.59
Soybean oil	0.80		
NaCl	0.60		
CaHPO_4_	0.10		
MgO	0.10		
Selenium yeast	0.02		
Premix	0.14		
Total	100.00		

**Table 3 animals-13-03523-t003:** Species abundance at the phylum level of calf gut microbes %.

Index	CON	OEO	CSB	EPS
Before weaning				
Firmicutes	57.29	56.97	54.17	57.19
Bacteroidota	38.77	37.49	43.06	39.24
Spriochatetota	1.93	3.56	0.86	1.38
Actinobacteriota	1.30	1.74	1.26	0.50
Firmicutes/Bacteroidota (F/B)	1.48	1.52	1.26	1.46
After weaning				
Firmicutes	55.64	57.99	47.63	57.19
Bacteroidota	39.93	36.43	46.32	38.39
Actinobacteriota	1.10	1.84	4.17	1.33
Spriochatetota	1.49	1.03	0.37	2.30
Proteobacteria	0.88	2.10	1.14	0.34
Firmicutes/Bacteroidota (F/B)	1.39	1.59	1.03	1.49

**Table 4 animals-13-03523-t004:** Species abundance at the genus level of calf gut microbes %.

Index	CON	OEO	CSB	EPS
Before weaning				
*norank_f__Muribaculaceae*	12.05	7.27	12.49	10.82
*Ruminococcus*	4.59	5.57	13.16	4.25
*unclassified_f__Lachnospiraceae*	6.06	7.66	2.50	5.62
*UCG-005*	6.92	3.61	4.31	6.41
*Prevotellaceae_NK3B31_group*	1.70	4.63	5.63	8.98
*Prevotella*	5.69	7.87	3.97	2.66
*Bacteroides*	5.25	5.74	5.08	1.81
*Rikenellaceae_RC9_gut_group*	5.02	2.12	5.71	3.55
*Faecalibacterium*	5.09	7.93	1.16	2.25
*Alloprevotella*	4.40	2.15	4.05	5.79
After weaning				
*norank_f__Muribaculaceae*	15.85	8.02	16.65	11.66
*UCG-005*	10.93	9.70	4.97	11.90
*Prevotellaceae_NK3B31_group*	2.74	11.50	6.37	7.08
*unclassified_f__Lachnospiraceae*	5.34	7.00	8.63	4.48
*Prevotella*	8.81	1.91	7.12	6.39
*Christensenellaceae_R7_group*	5.55	10.01	0.63	0.60
*Ruminococcus*	2.29	2.61	4.24	6.13
*Rikenellaceae_RC9_gut_group*	3.04	2.15	5.19	2.99
*Alloprevotella*	3.53	5.42	1.78	2.54
*Blautia*	4.59	2.58	3.07	2.05

**Table 5 animals-13-03523-t005:** Functional abundance of gut microbes in calves (in the KEGG level 2 pathway) %.

Index	Before Weaning				After Weaning			
	CON	OEO	CSB	EPS	CON	OEO	CSB	EPS
Global and overview maps	40.83	40.71	40.99	40.60	40.76	40.64	40.83	40.69
Carbohydrate metabolism	9.84	9.87	9.99	9.67	9.81	9.66	9.92	9.80
Amino acid metabolism	7.16	7.05	7.15	7.09	7.22	7.15	7.12	7.13
Energy metabolism	4.25	4.20	4.29	4.27	4.22	4.28	4.25	4.26
Metabolism of cofactors and vitamin	4.16	4.22	4.25	4.24	4.18	4.31	4.22	4.19
Translation	3.77	3.74	3.83	3.90	3.85	3.90	3.76	3.81
Replication and repair	3.23	3.24	3.29	3.35	3.21	3.32	3.23	3.27
Membrane transport	3.07	3.09	2.26	2.91	3.03	2.78	3.02	2.99
Nucleotide metabolism	2.87	2.85	2.89	2.96	2.88	2.93	2.88	2.87
Signal transduction	2.17	2.19	2.13	2.16	2.17	2.17	2.07	2.20

**Table 6 animals-13-03523-t006:** Analysis results of volatile fatty acids in the blood of calves (ng/mL).

Index	CON	OEO	CSB	EPS	SEM	*p*-Value
Acetic acid	355.40 ± 60.46 ^b^	462.95 ± 72.29 ^a^	418.36 ± 86.84 ^a^	453.05 ± 83.73 ^a^	11.248	0.002
Propanoic acid	147.60 ± 51.01 ^ab^	172.43 ± 52.47 ^ab^	189.46 ± 57.49 ^a^	135.82 ± 62.02 ^b^	7.662	0.054
Isobutyric acid	2.49 ± 0.78	3.13 ± 1.56	2.43 ± 0.95	2.74 ± 0.82	0.143	0.286
Butyric acid	153.93 ± 70.56	241.90 ± 287.57	139.35 ± 83.89	122.87 ± 83.48	21.560	0.201
Valeric acid	3.14 ± 1.02 ^b^	4.99 ± 2.60 ^a^	4.86 ± 1.64 ^a^	4.60 ± 1.92 ^a^	0.261	0.044
Isovaleric acid	2.95 ± 1.35	3.97 ± 1.14	3.80 ± 2.42	3.94 ± 1.71	0.228	0.363

Different superscript lower-case letters a and b indicate a significant difference at *p* < 0.05.

## Data Availability

Follow-up research on this project is ongoing; please contact the corresponding author with reasonable requests.

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
