# Peer review of "Effects of Essential Oil and/or Encapsulated Butyrate on Fecal Microflora in Neonatal Holstein Calves"

_animals, 2023, doi:10.3390/ani13223523_

Round 1

Reviewer 1 Report

Comments and Suggestions for Authors

It is widely recognized that early weaning of the newborn calf depends on good GI tract development. This paper will contribute to accumulating scientific knowledge on the effect of essential oil, butyrate, and mixture as an additive to improve health the neonatal calves. As shown below, there are several unclear parts in the manuscript.

1. Consideration of the scientific names for microbes and the sentient spacebar was almost entirely absent from the abstract and introduction sections.

2. Material and methods part,

2.1 Even though the effects of sex or birthdate may be related, the author decided which newborn calf would get each treatment based on the animal unit. The author did not evaluate the amount of milk consumed, which might affect how well the treatment is effective, nor did they describe the overall health of the calf.

2.2 The serum's volatile fatty acid content was determined, and the author needed to describe the methodology.

2.3 The experiment was carried out to evaluate the effects of feeding an essential oil, butyrate, and its mixture to dairy calve. Since your research focuses primarily on the combined effect, I am concerned about your experimental design because it does not specify how the treatment group's hypotheses would be contrasted.

3. The discussion and result sections need to be thoroughly reevaluated in order to revise the milk consumption parameter and overall health status.

4. The references section should be written in accordance with journal formatting instructions.

Author Response

Reviewer 1

Dear reviewers:

We feel great thanks for your professional review work on our article.As you are concemed, there are several problems that need to be addressed.According to your nice suggestions, we have made extensivecorrections to our previous draft, the detailed corrections are listed below.

  1. Consideration of the scientific names for microbes and the sentient spacebar was almost entirely absent from the abstract and introduction sections.

Response:Thank you for your valuable comments, I apologize for our carelessness, thanks for the reminder, I have made changes in the abstract and introduction section, thanks again.

  1. Material and methods part,
    • 1 Even though the effects of sex or birthdate may be related, the author decided which newborn calf would get each treatment based on the animal unit. The author did not evaluate the amount of milk consumed, which might affect how well the treatment is effective, nor did they describe the overall health of the calf.

Response:Thank you for your comments. When we conducted the trial, we purposely excluded underweight, sick at birth, or weak calves, and all calves enrolled in the experimental group were healthy individuals free of diseases such as pneumonia and weighing ≥35 kg. Throughout the trial period, as the calves grow, there is a fixed number of times and amount of milk feeding at each stage, this condition is maintained, which was not described clearly in the text before, I have added in the text, thank you!(L126-134)

2.2   The serum's volatile fatty acid content was determined, and the author needed to describe the methodology.

Response:Thank you for pointing this out and again I apologize for my carelessness, I have added the equipment and methods used for volatile fatty acid determination in the Materials and Methods section of the text.(L174-176,L236-242)

2.3   The experiment was carried out to evaluate the effects of feeding an essential oil, butyrate, and its mixture to dairy calve. Since your research focuses primarily on the combined effect, I am concerned about your experimental design because it does not specify how the treatment group's hypotheses would be contrasted.

Response:Thanks for your valuable advice, we are trying to study the microbiological effects of oregano essential oil and sodium butyrate fed to calves, and whether this effect can lead to better growth performance and health status. In addition, we would like to explore whether the two substances are added together to produce synergistic or antagonistic effects, and compare between treatment groups. Mainly by assessing which microorganisms were changed by these two substances and what impact these microorganisms brought to calves, I revised the purpose, results and discussion part of the experiment in this paper to strengthen the discussion on this issue, and I also hope to get your opinions. I will continue to improve this issue, thank you very much!

  1. The discussion and result sections need to be thoroughly reevaluated in order to revise the milk consumption parameter and overall health status.

Response: Thanks for your suggestion, we have resorted some of the results, added a discussion on the ratio of Bacteroides and firmicutes, added some references in the discussion part to support our results, and supplemented the experimental design part. If there is any problem, please point out that I will continue to improve, thank you very much for your suggestion. (L559-589)

  1. The references section should be written in accordance with journal formatting instructions.

Response: Thanks for the heads up, I've made the changes in the references section.

Reviewer 2 Report

Comments and Suggestions for Authors

Review report

animals-2692958

Effects of Essential Oil and/or Encapsulated Butyrate on Fecal 2 Microflora in Neonatal Holstein Calves

Zhihao Luo 1, Ting Liu11*, Peng Li 1, Shuru Cheng1, David. P. Casper2,3

1Faculty of Animal Science and Technology, Gansu Agricultural University, No. 1 Yingmen Village Anning, 5 Lanzhou, 730070, China.

2Casper’s Calf Ranch, 4890 West Lily Creek Road, Freeport, IL 61032, USA

3Department of Animal Sciences, North Carolina A&T State University, Greensboro, NC 27411

*Corresponding author: [email protected]

Summary Rating

Originality:                             fair

Technical Quality:                  fair to good

Clarity of Presentation:           fair to good

Importance to field:                fair

Evaluation:

This manuscript will be acceptable after revisions

Comments:

The manuscript aims to investigate the effects of feeding oregano essential oil, butyr-19 ate and its mixture on intestinal microbial diversity of calves. Authors concluded that the addition of oregano essential oil, butyrate and their mixtures to milk fed to calves can regulate the microbial structure. Although the originality is not high, the work is useful locally in the feeds and ruminant nutrition areas. The technical quality and clarity of presentation are fair to good. The manuscript is well-written.  But they are some shortcomings in this study.  Please see the suggestion as follow. I recommend that this manuscript will be acceptable after revision.

Please see the following for my suggestions to be considered:

·       First, prevent editorial errors in the paper. I suggest the authors work with an individual who is proficient in English. Have the individual provide a careful and thorough review of the manuscript for grammatical issues (complete sentences, tense, etc.) to help ensure the authors thoughts are clearly transmitted to the reader. 

·       In statistical analysis section.: You used CRD model to analysis parameter such as VFA data(table3). Have you checked CRD model assumption (NIID) in your data? Please do residual analysis to check your model assumption. Also tell us your final model, common variance model or unequal variance model for final analysis.

·       Table 3: add SEM and P value

It is an interesting research project.

Comments on the Quality of English Language

see the above for the review report

Author Response

Dear reviewers: We feel great thanks for your professional review work on our article.As you are concemed, there are several problems that need to be addressed.According to your nice suggestions, we have made extensivecorrections to our previous draft, the detailed corrections are listed below.

  1. First, prevent editorial errors in the paper. I suggest the authors work with an individual who is proficient in English. Have the individual provide a careful and thorough review of the manuscript for grammatical issues (complete sentences, tense, etc.) to help ensure the authors thoughts are clearly transmitted to the reader. 

Response:Thanks for your suggestion. We have tried our best to polish thelanguage in the revised manuscript. English is not my native language, so if there are still language errors in the article, please point them out and I will continue to make corrections.

  1. In statistical analysis section.: You used CRD model to analysis parameter such as VFA data(table3). Have you checked CRD model assumption (NIID) in your data? Please do residual analysis to check your model assumption. Also tell us your final model, common variance model or unequal variance model for final analysis.

Response:We used SPSS software to conduct one-way ANOVA for data analysis of volatile fatty acids, with different treatments as variable factors. We are sorry that we did not explain the analysis method of VFA data clearly in the previous writing due to our carelessness, and now we have added it in the paper. Thank you again for your reminding!(L257-259)

  1. Table 3: add SEM and P value

Response:Thank you for pointing this out, it is important for me to refine the completeness and readability of the data, due to changes elsewhere in the text, the serial number of Table 3 was changed to Table 6, and I have added the corresponding SEM and P values to Table 6, thank you for the reminder!(Table6-L418)

Reviewer 3 Report

Comments and Suggestions for Authors

Ms. Ref. No.:  animals-2692958

Title: “Effects of Essential Oil and/or Encapsulated Butyrate on Fecal Microflora in Neonatal Holstein Calves

Animals

General comments

I have had the opportunity to review the manuscript. The manuscript is interesting and is in the topic of the journal, however it needs more details in the methodology some points in the discussions

Below my considerations:

Introduction

L 60 ….ruminococcus… capital letter

L 63-63 need add references

L 70-72 need add reference

L 89-96 need add rederences

An explicit definition of the purpose of the study is missing, it must be added.

Material and methods

L 118-121 it is necessary to rephrase the sentence to make it clearer, there is some incorrect punctuation and the sentence as a whole is not clear.

L 136 “The feed was fed…..” reformulate

It is necessary to add the analytical composition of the essential oils administered.

L 211 it is not correct to use the wording …etc…

It is necessary to specify the statistical model used, which factors are fixed and which are variable.

Results

It is preferable to show the results of the numerous percentages in the table, they cannot be followed within the manuscript.

Author Response

Dear reviewers:

We feel great thanks for your professional review work on our article.As you are concemed, there are several problems that need to be addressed.According to your nice suggestions, we have made extensivecorrections to our previous draft, the detailed corrections are listed below.

  1. Introduction:L 60 ….ruminococcus… capital letter,L 63-63 need add references,L 70-72 need add reference,L 89-96 need add rederences,An explicit definition of the purpose of the study is missing, it must be added.

Response: As you suggested, we have added references in the introduction section to support the issues told in the text(reference:7,8,9,10,17,18,19), and summarized and modified the purpose of the study. (L112-115) The correct writing of the microbiological names in the introduction has also been revised, thank you!

  1. Material and methods:L 118-121 it is necessary to rephrase the sentence to make it clearer, there is some incorrect punctuation and the sentence as a whole is not clear.L 136 “The feed was fed…..” reformulateIt is necessary to add the analytical composition of the essential oils administered.L 211 it is not correct to use the wording …etc…,It is necessary to specify the statistical model used, which factors are fixed and which are variable.

Response: Thank you for pointing out this problem, which I think is very crucial for me to improve the deficiencies in the paper. I have made the selection conditions of experimental animals in the part of the paper, supplemented the feeding procedures, added the ingredients of additives used, and explained the fixed conditions in the part of data processing. If there are still mistakes, please point out, I will continue to improve, thank you very much for your advice and help.

  1. Results:It is preferable to show the results of the numerous percentages in the table, they cannot be followed within the manuscript.

Response: Thank you for your suggestion, it will indeed become more intuitive and clearer to present the results of this section in a table, I have modified the article by making the percentage results of the three sections in Table 3, Table 4 and Table 5 respectively and added a ratio of anaplasmosis and thickened wall bacteria in Table 3, which is more appropriate to complement the results, I sincerely thank you for your valuable suggestion, thank you!

Round 2

Reviewer 3 Report

Comments and Suggestions for Authors

ok

Comments on the Quality of English Language

ok